# Reversing the Decline in Threatened Species through Effective Conservation Planning

**Onnie Byers** *[ID], **Jamieson Copsey, Caroline Lees** [ID]**, Philip Miller and Kathy Traylor-Holzer** [ID]

IUCN Species Survival Commission Conservation Planning Specialist Group, 12101 Johnny Cake Ridge Road, St. Paul, MN 55124, USA
* Correspondence: onnie@cpsg.org

**Abstract:** Despite the committed action by many in past decades, recent reviews show little progress in slowing species declines, and future waves of extinction are predicted. Not only do such declines signal a failure to meet international commitments to stem biodiversity loss and undermine the potential for achievement of the species-related target in the Post-2020 Biodiversity Framework, but they also jeopardize our ability to achieve the 2030 Sustainable Development Goals, many of which rely on the resources provided by species and the ecosystems they support. A substantial increase in ambition and the application of tools at the global scale and across all elements of the species conservation cycle—Assess, Plan, and Act—is urgently needed to create swift and lasting positive change for species. Well-resourced, effectively implemented species conservation plans play a key role in meeting this challenge. Here, the IUCN SSC Conservation Planning Specialist Group (CPSG) presents a proven approach to species conservation planning that emphasizes the thoughtful design and facilitation of collaborative processes that feature the rigorous scientific analysis of quantitative data on species biology and impacts of anthropogenic threats and their mitigation through management. When incorporated from the beginning of a species management project, the CPSG's principles and steps for conservation planning can help reverse the decline of threatened species.

**Keywords:** One Plan Approach; Conservation Planning Specialist Group; species conservation planning; threatened species; IUCN; Species Survival Commission





## 1. Introduction

Today, people share the Earth with an estimated 8.7 million species [1]. Of the more than 2 million that have been identified and described, only 147,517 species (<8%) have been assessed for the IUCN Red List of Threatened Species [2], which is the world's leading authority on the conservation status of species. Of these, 41,459 species are considered Critically Endangered, Endangered, or Vulnerable, meaning they are threatened with extinction [2]. Despite committed action by many in past decades, recent reviews show little progress on slowing declines, and future waves of extinction are predicted [3,4].

Not only do such declines signal a failure to meet international commitments to stem biodiversity loss [4] and undermine the potential for achievement of the species-related targets in the Post-2020 Biodiversity Framework [5], they also jeopardize our ability to achieve the 2030 Sustainable Development Goals [6], many of which rely on the resources provided by species and the ecosystems they support. There are, however, data that illustrate that without the conservation actions of the past decades, the situation would be far worse [7]. We know that conservation action works, but a substantial change in approach and ambition is needed to generate increased momentum against the rising tide of extinction and to create lasting positive change for species. As an early step toward achieving this goal, this paper broadly describes the work of the SSC's Conservation Planning Specialist Group (CPSG) and its approach to the design and facilitation of evidence-based, collaborative

planning processes for species conservation. The organization's principles and steps for effective conservation planning are described, emphasizing the measurable improvements in species status that are possible when this approach is adopted by planning authorities. New tools are described that both expand our ability to evaluate conservation needs for more endangered taxa and improve the products of those assessments. A substantial upscaling of the production of well-resourced, collaboratively developed, and effectively implemented species conservation plans should be a cornerstone of future biodiversity conservation efforts worldwide.

## 2. Species Conservation Planning

Many species can be effectively conserved through general nature conservation measures such as: protecting and managing networks of protected areas, creating and enforcing laws on use and trade, protecting and restoring ecosystems or managing invasive species and disease outbreaks [8]. Relying solely on generalized measures, however, means that many species will not be adequately conserved, either because they have geographical ranges outside of protected areas, or because they have complex or idiosyncratic characteristics and needs, or because their viability is already so compromised that they may become extinct before these measures can take effect. For these species—and there are many thousands of them [8]—species-targeted planning is required.

Species conservation planning aims to increase the implementation and effectiveness of actions designed to reduce the risk of species decline and extinction. This is best accomplished by ensuring that the products of that planning: (i) are based on a thorough analysis of the best available information, (ii) include well-defined and achievable goals, (iii) incorporate the full range of stakeholder perspectives, (iv) foster agreement among those involved about what should be done to improve species status, and (v) include a timeline for review. The IUCN Species Survival Commission (SSC) recognizes the value of thoughtful, collaborative conservation planning processes, and it includes planning as one of the three elements of its Species Conservation Cycle: Assess–Plan–Act [9]. Within the IUCN SSC, species conservation planning is led and supported through the Conservation Planning Specialist Group (CPSG), and it is skillfully conducted by many other SSC specialist groups (e.g., Cat Specialist Group, Crop Wild Relatives Specialist Group, Primate Specialist Group) and partner organizations (e.g., BirdLife International and Botanic Gardens Conservation International).

The CPSG's signature planning process has typically focused on developing a detailed conservation action plan for a single species. These intensive planning exercises lead to a practical blueprint of actions designed to mitigate biological and anthropogenic threats to population persistence as well as address diverse social and institutional challenges to achieving those actions. The CPSG's approach is particularly useful for species whose conservation involves competing interests among multiple stakeholder groups and in planning contexts with high levels of uncertainty and complexity. CPSG-led workshops typically feature the scientific rigor of a population viability analysis (PVA) that helps wildlife biologists and managers more clearly understand the threats that influence a population, identify critical knowledge gaps, and evaluate potential actions [10]. PVA is combined with tools for helping people organize and evaluate information across a broad range of disciplines and perspectives to identify problems, common goals, and potential solutions. Through this integration, planning workshop participants create conservation actions for species that also take into account the social, cultural, and economic needs of local people. When all stakeholders participate as active and equal contributors in building the plan, they are much more likely to support its implementation [11–13].

## 3. SSC's Species Conservation Planning Principles and Steps

The CPSG's approach, adopted by the SSC, is described by a series of eight planning steps implemented using seven guiding principles. Taken together, these principles and steps are important elements in the development and implementation of effective species

conservation plans (Figure 1) [14]. The principles and, with slight modification, the steps work well for both single and multi-species planning.

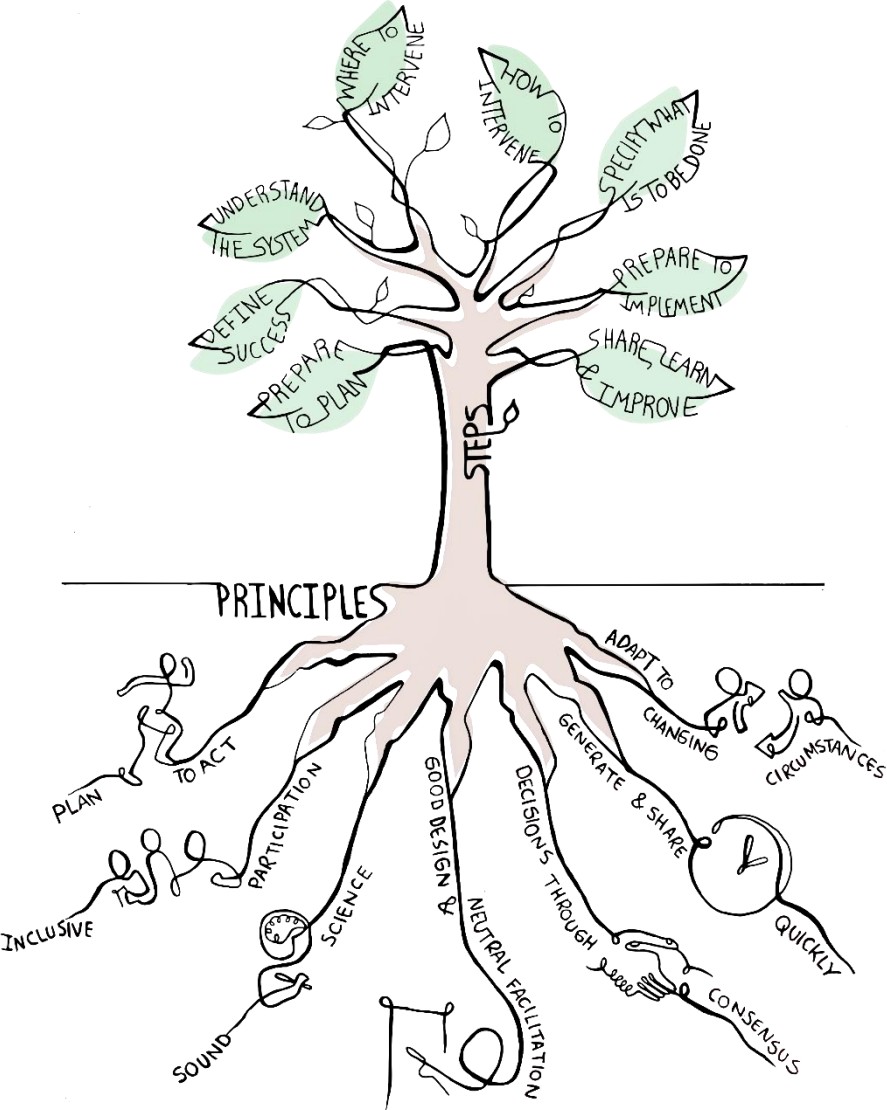

**Figure 1.** The SSC CPSG approach to species conservation planning: Core principles are represented as stable roots, while the leaves represent the planning steps that continue to evolve in response to the increasing complexity of today's wildlife conservation challenges. Design and artwork: Eugenia Cordero Schmidt.

### 3.1. Principles

For 40 years, the CPSG has been assisting diverse groups to plan for the conservation of species. Its approach to planning is deeply rooted in a set of principles that emphasize sound science and the meaningful participation of key stakeholders. These principles (Table 1) are used to guide the series of planning steps that continue to evolve in response to the increasing complexity of today's wildlife conservation challenges.

**Table 1.** SSC CPSG Species Conservation Planning Principles.

| | |
|---|---|
| Plan to act | The intent of planning is to promote and guide effective action to save species. This principle underpins everything we do. |
| Promote inclusive participation | People with relevant knowledge, those who direct conservation action, and those who are affected by that action are all key to defining conservation challenges and deciding how those challenges will be addressed. Inclusivity refers not only to who is included in the planning process but also to how their voices are valued and incorporated. |
| Use sound science | Working from the best available information—whether that be established facts, well-supported assumptions or informed judgments—is crucial to good conservation planning. Using science-based approaches to integrate, analyze, and evaluate this information supports effective decision making. |
| Ensure good design and neutral facilitation | Good species planning is designed to move diverse groups of people through a structured conversation in a way that supports them to coalesce around a common vision for the species and to transform this into an achievable, effective plan. Facilitators skilled in planning are essential in guiding these processes. Critically, neutral facilitation eliminates potential or perceived bias in the planning process, helping participants to contribute their ideas and perspectives freely and equally. |
| Reach decisions through consensus | Effective species conservation planning results in decisions that all participants can support or accept. Recognizing shared goals, seeing the perspective of others, and proceeding by consensus helps galvanize participants behind a single plan of action that is more likely to be implemented. |
| Generate and share products quickly | Producing and sharing the products of a conservation planning process quickly, freely and widely are important factors in its success. Delays carry a cost in terms of lost momentum, duplicated or conflicting effort or missed opportunities for action. |
| Adapt to changing circumstances | Effective plans are those that evolve in response to new information and to changing circumstances—biological, political, socio-economic, and cultural—that influence conservation efforts. Plans are considered living documents that are reviewed, updated, and improved over time. |

The principles that guide CPSG's approach to planning are depicted as roots of the planning tree to emphasize their permanent nature and the important foundation they provide. Well-designed and well-executed species conservation planning that adheres to these seven core principles can improve existing efforts and stimulate greater ambition, collaboration, and resourcing. While planning steps may differ, CPSG has found these principles to be essential conditions for success.

Underpinning these principles is a commitment to the One Plan Approach [15]: the collaborative development of management strategies and conservation actions by all responsible parties to produce one comprehensive conservation plan for the species, whether inside or outside its natural range. The result is an integrated conservation plan that mobilizes the full suite of skills, expertise and resources available to species in trouble, giving them a better chance at a future in the wild. Application of the One Plan Approach has continued to grow since being defined by the CPSG in 2011 [15]. In 2014, this approach was a key conceptual framework that informed the development of the IUCN SSC's Guidelines on the Use of Ex Situ Management for Species Conservation [16], and in 2020, IUCN members supported Resolution 094, which urged the IUCN Secretariat and professional societies to promote the integration of in situ and ex situ conservation interventions by applying the One Plan Approach [17].

*3.2. Steps*

The CPSG's planning approach is made up of eight steps (Table 2). These steps are flexible, and individual practitioners may use different terminology or merge certain steps [18,19].

**Table 2.** SSC CPSG Species Conservation Planning Steps.

| | |
|---|---|
| Prepare to plan | Agree on the scope, rationale, and required product of planning. Design and prepare a planning process that will meet these requirements. |
| Define success | Define the core elements of a future state for the species that represents the desired outcome both for conservation and for other relevant stakeholder needs or values. |
| Understand the system | Assemble the best available information on the biology, history, management, status and threats to the species, the obstacles to addressing those threats, and the opportunities or options for successful intervention. |
| Decide where to intervene | Determine where in the system to intervene, recommend, and prioritize the changes needed to achieve the desired future state. |
| Agree on how to intervene | Identify alternative approaches to achieving the recommended changes, compare their relative costs, benefits and feasibility, and choose which one(s) to pursue. |
| Specify what is to be done | Agree on what will be done, when and by whom, to implement the chosen approach, and which measures will be used to indicate progress or completion of specific tasks. |
| Prepare to implement | Agree on how key individuals and organizations will communicate, coordinate, make decisions, and track and report on progress as they move forward together to implement the plan. |
| Share, learn and improve | Produce the plan swiftly, share it widely and strategically to maximize conservation impact, and capture lessons learned in order to develop more effective conservation planning processes. |

While there are, of course, variations among the various planning approaches, for the most part, the steps (and, in fact, those of most planning cycles regardless of industry) are generally the same. The processes all contain a needs assessment or problem formulation, information gathering and analysis, establishment of goals and objectives, identification and evaluation of alternative actions, and preparation for implementation of the chosen alternative(s).

*3.3. Implementation*

The first of the CPSG's planning principles is "Plan to Act", recognizing that there is little value to planning if it does not lead to action. Although action follows planning in typical project management cycles (and in the SSC's Assess–Plan–Act species conservation cycle), the two should not be discrete. Good planning should assist the management, monitoring, and evaluation of implementation.

The CPSG's principles and steps approach aims to support the implementation of recommended actions resulting from a planning project by: (1) considering who will or could implement the resulting plan and who might hamper implementation, before planning begins, so that wherever possible, these stakeholders are engaged in the process, (2) aiming to develop goals and objectives that are clearly described, achievable, well-targeted, and that make clear how progress will be measured, (3) explicitly considering the organizational structure of the implementation project, how it will be managed and communicated, and how decisions will be made, (4) investing time and energy in stakeholder consensus building to enable a form of environmental and intergenerational justice [20] and to give implementation a solid platform, and (5) setting a time-frame for review and re-planning, recognizing that the first plan will not be perfect and may need to be revised. Setting an expectation of periodic re-evaluation from the outset that incorporates new information and experience creates good conditions for project evolution.

**4. Measuring Impact on Reversing Species Declines**

This style of planning has been shown to provide a turning point for those involved in conserving species, helping them transition to more effective ways of collaborating [21]. Over time, this leads to clear and measurable improvements in species' conservation status, as shown in a new study comparing the extinction trends of species before and after a planning intervention [13].

Researchers drew from a CPSG in-house database of all species-level planning projects that took place before 2008 (to allow for at least 10 years of post-planning changes) and

for which the species involved had been assessed for the IUCN Red List multiple times and, importantly, both before and after the CPSG conservation planning workshop. For the 45 species projects that met the criteria, an aggregate group extinction trajectory before and after planning was calculated (for further details on methodology and data sets, see Lees et al., 2021). This took the shape of a steep decline before the planning workshop, which was followed by a period of continued but shallower decline after the workshop, culminating in an upturn within 15 years. None of these species with planning went extinct. For comparison, a "without planning" trajectory was simulated for these same species, based on patterns of change in extinction risk prior to planning. This trajectory declined throughout the post-planning period and resulted in the extinction of around eight species. The difference between these two trajectories was statistically significant at both the 10- and 15-year marks (Figure 2).

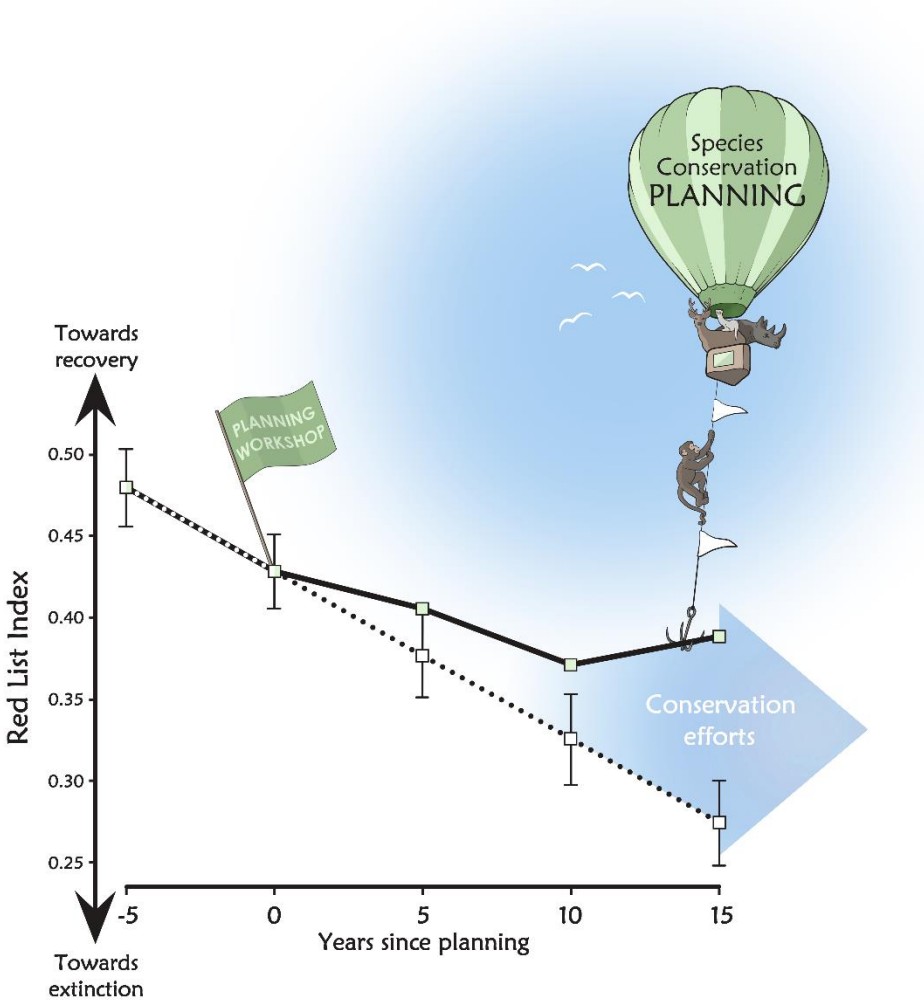

**Figure 2.** CPSG-style planning helps to reverse the decline of threatened species. The Red List Index (RLI) uses IUCN Red List categories to measure the projected overall extinction risk over time. An RLI value of 1.0 equates to all species being categorized as 'Least Concern', and hence that none are expected to go extinct in the near future. A value of 0 indicates that all species have gone extinct. Graphic adapted from Lees et al., 2021 [13].

Although the study sample is small, it is grounds for optimism, especially as the study set was dominated by larger-bodied, longer-lived taxa with small or fragmented populations as well as other challenging threats, such that longer times to recovery might normally be predicted. Many years were required to accumulate the data for this study, and

the CPSG will continue to expand the sample in the years to come. Nevertheless, the results to date are clear and are consistent with the view that CPSG-style conservation planning can be a turning point for a species, helping to reverse the decline in threatened species.

## 5. Effective Planning for More Species: Tools, Processes, Capacity and Funding

Intensive planning for individual species can lead to highly effective outcomes [13]; however, this approach alone is insufficient to address the global biodiversity crisis [8]. While the exact number of threatened species that are currently covered by a conservation plan is unknown, it is clear that the vast majority are not the subject of logically structured, evidence-based, stakeholder-inclusive conservation plans. With more than 40,000 species categorized by the IUCN Red List as at risk for extinction, the conservation community must urgently prioritize and improve efficiency in its conservation planning efforts and continue to upscale planning capacity to ensure that threatened species in need are covered by effective, implemented conservation plans. To date, the planning work of the CPSG and of the wider SSC has impacted hundreds of species, and other agencies around the world are planning for the conservation of thousands more. Unfortunately, the number still in need of plans is huge and expected to grow. In recognition of this, the SSC has set itself the challenge of ensuring that every species that needs a plan is covered by an effective plan.

Although intensive planning processes for individual species will remain an important component of SSC's future efforts, multi-species planning methodologies are required to address a larger number of threatened taxa more efficiently. Past studies indicate that in general, species covered only within multi-species plans are less likely to exhibit improving status trends than those with their own individualized plan [11]; however, the scale and urgency of the need, along with the shortfall in currently available resources, requires greater efficiency in planning. One solution is to group species for multi-species planning based on well-chosen groupings that promote more effective planning.

The CPSG's Assess-to-Plan (A2P) process and associated tools are designed to support the rapid identification of multi-species groups for planning and action as well as potential champions able and willing to take these recommendations forward [22,23]. Effective themes for grouping species for further multi-species planning generally coalesce around those described in Figure 3 [24–30], recognizing that a single species may fall into more than one grouping.

These groups typically center around geography or physical environment (e.g., sites, areas, habitats) or around specific threats (e.g., disease, illegal trade) that could be the basis of further planning. In addition, some species may require intensive population management in addition to threat abatement and may be candidates for the CPSG's Ex situ Conservation Assessment (ECA) process. During the A2P process, some species may be identified as requiring detailed individual planning and cannot or should not be grouped for multi-species planning.

Addressing the SSC's goal to ensure that every species that needs a plan is covered by an effective plan will require: (1) rapid identification of threatened species that are not adequately covered by plans, (2) the advancement of larger numbers of species from status assessment into conservation action through effective multi-species planning, (3) massive expansion of the capacity to build effective plans, and (4) sufficient funding to implement the above. Below, we offer suggestions to address these needs.

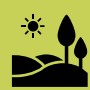

**Site-specific** (a few specific, identifiable sites are key – e.g. Key Biodiversity Areas, Alliance for Zero Extinction sites - and challenges are local to those site)

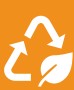

**Ecosystem/habitat specific** (threats travel with the ecosystem/habitat on which the species relies – **e.g. mangroves, very old trees and their features**)

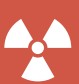

**Threat-specific** (threats travel with the species – illegal trade, disease, human-wildlife conflict, invasive species)

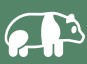

**Single-species** (umbrellas or flagships, or species with complex and non-overlapping issues)

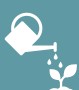

**Intensive care** (threat abatement insufficient – extra support needed, may require *ex situ* support)

**Figure 3.** In an Assess-to-Plan (A2P) process, multi-species groups are rapidly identified for planning and action. Effective groupings generally coalesce around these themes, recognizing that a single species may fall into more than one grouping.

*5.1. Rapid Identification of Species Needing Planning*

The Red List assessment process offers an ideal opportunity to systematically collect, and make available, information on planning coverage across threatened species. A slight change in instructions to assessors and the identification of a searchable field in which to capture information on the existence of species plans in Species Information Service (SIS), the IUCN's web application for conducting and managing species assessments for the IUCN Red List, would go a very long way to helping to overcome this obstacle. In addition, large databases may be valuable in identifying gaps and opportunities for planning [31].

*5.2. Effective Multi-Species Planning*

The Assess-to-Plan (A2P) process is proving to be a valuable tool, particularly for specious taxonomic groups or geographic areas. Combined with the Red List assessment process, A2P allows more species to move more quickly from assessment to conservation action. It uses Red List data to group species according to overlapping conservation needs. Key to the success of this process is close collaboration between Red List assessors and planning facilitators from the outset of the Red List assessment process to ensure that the right information is captured and in ways that support subsequent retrieval and analysis for this purpose. Plans are underway to automate components of the process to reduce the personnel time and training required to conduct A2P so it can be shared and used widely. Note that for species falling into the A2P "Intensive Care" grouping, a multi-species Ex

situ Conservation Assessment can be conducted to evaluate the conservation value and feasibility of various ex situ activities [32,33].

*5.3. Expansion of Planning Capacity*

Developing this capacity globally is achievable but will require extensive training, mentoring, coaching and support. If each national government, IUCN SSC Specialist Group, interested non-government organization, zoo, aquarium, botanic garden, and civil society group with a concern for species conservation had within it a cadre of competent planners able to respond as required, there would be more than sufficient capacity to meet the planning need.

To build such coordinated global capacity for saving biodiversity demands a shared understanding of what effective species conservation planning looks like. The CPSG has developed the principles and steps described above with this in mind. They present a succinct philosophy and framework for good planning based on four decades of evolving practice.

Since 2018, 1153 individuals from 92 countries have completed training courses in the CPSG's conservation planning principles, steps, tools and methodologies. SSC Specialist Group members, government representatives, and zoo, aquaria, and botanic garden staff have been the focal audience to date, with priority given to individuals already actively engaged or responsible for developing species plans. A recent post-training survey showed that participants who have completed the CPSG's training program feel more capable, confident, and motivated to undertake species conservation planning work [34]. Half of the survey respondents were involved in designing and/or facilitating one or more species conservation planning processes in 2020, encompassing several hundred species in countries around the world.

*5.4. Sufficient Funding*

An enormous gap exists between the amount of funding needed for biodiversity conservation and the amount devoted to it. While a 2014 global report estimated that the investment required is between US$130 and $440 billion annually [35], currently available data indicate that global biodiversity finance, the practice of raising and managing capital to support and conserve biodiversity [36], is estimated at US$78–91 billion per year [37]. Species planning and the implementation of resulting plans are frequently identified as meaningful conservation measures to which sufficient resources must be allocated [38,39]. The IUCN, Species Survival Commission, key partners, and the biodiversity-related conventions are developing the Global Species Action Plan (GSAP) to support implementation of the Post-2020 Global Biodiversity Framework [40]. The GSAP outlines the actions required under all the proposed targets in the Post-2020 Global Biodiversity Framework in order to conserve species effectively. The GSAP will provide a toolkit of resources to assist governments and other stakeholders in implementing actions to conserve species. In addition, the GSAP calls on governments, investors, and financial institutions to "ensure that financial flows and development financing is based on safeguards ensuring positive impacts on threatened species and critical habitats" and donors and the philanthropic community to "increase substantially the resources invested in conservation and the sustainable use of species and support innovative mechanisms for financing species conservation". Species conservation planning tools and capacity-building resources are a vital component of this first-of-its-kind global program of work for species.

## 6. Discussion

In general, species are conserved through the effective protection and management of representative networks of natural areas, in combination with the systematic prevention and mitigation of threats that operate within and outside these areas. However, in the current reality, these measures may not be enacted with either sufficient urgency or scope to reverse the declines of tens of thousands of species already threatened with extinction. To add another layer of complexity to this reality, many conservation activities will need

to be tailored to the specific needs of taxonomic groups that differ in their habitat requirements, demographics and life histories. Consequently, dedicated planning to address these complexities is a necessity.

While the problems facing biodiversity are clearly global in scope, and there is tremendous value in aggregating information to provide an indication of global trends, particularly in terms of status assessment, it is often at a national or subnational level where the planning and acting elements of the SSC's species conservation cycle are most appropriately applied [41]. As the distribution of threatened species often extends beyond administrative boundaries, we must refine our approach to planning at the appropriate spatial scale to achieve the best outcomes for species stability. A new initiative within the SSC known as Reverse the Red [9], in collaboration with the national level Centers for Species Survival [42], is dedicated to applying the Assess–Plan–Act framework at the national level, empowering governments to efficiently implement key elements of a wide-ranging species conservation plan at the appropriate scale for greater impact. In addition, the IUCN's Global Species Action Plan framework is designed to support countries in their efforts to achieve relevant targets set forth in the Post-2020 Biodiversity Framework [5]. These targets include those that will explicitly require application of elements of the Assess–Plan–Act cycle as well as the adoption of a One Plan Approach for the effective engagement of a broad range of key experts and stakeholders.

There is an urgent need to upscale global capacity to ensure that every species that needs a plan is covered by a well-resourced and effectively implemented plan. Integrated within the broader IUCN Global Species Action Plan, the CPSG's principles and steps for species conservation planning, in coordination with associated training and mentoring programs, are providing a pathway to achieving this capacity. If upscaled to a level appropriate to respond to the need, the implementation of the resulting plans will help to reverse the decline in threatened species.

**Author Contributions:** Conceptualization, O.B.; writing—original draft preparation, O.B.; writing—review and editing, O.B., C.L., P.M., K.T.-H. and J.C. All authors have read and agreed to the published version of the manuscript.

**Funding:** This manuscript is one of the CPSG activities supported by the Global Conservation Network.

**Institutional Review Board Statement:** Not applicable.

**Data Availability Statement:** Not applicable.

**Acknowledgments:** We are grateful to the Global Conservation Network for the support of CPSG activities.

**Conflicts of Interest:** The authors declare no conflict of interest. The funders had no role in the design of the study; in the collection, analyses, or interpretation of data; in the writing of the manuscript; or in the decision to publish the results.

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
