# Peer review of "Reversing the Decline in Threatened Species through Effective Conservation Planning"

_diversity, doi:10.3390/d14090754_

Round 1

Reviewer 1 Report

General Comments

This is a well-written outline of the approach to conservation planning taken by the IUCN CPSG. I think it will be a useful document for those interested in conservation planning (which should be everyone) for many years to come.

One element that I feel is missing is a clear statement at the end of the introduction to advise the reader on what this manuscript is about and what it hopes to achieve. The last sentence of the abstract sort of does this but a similar more detailed statement is needed at the end of the introduction. That statement at the end of the abstract falls a bit short of what I think should be covered in an abstract. Another sentence or two should better outline what is covered. I like that the abstract is very concise but there is not enough meat on the last sentence for someone reading just the abstract to want to read further.

Specific Comments

L57. What is ‘good’ information? Use a better term. Detailed? Reliable? Available?

L59. And should include a timeline for review. See L168-170.

Fig. 2. Define the Red List index values.

L225. Reference 8 doesn’t look relevant at that point but perhaps at the end of the sentence.

Fig. 3. Define ‘KBAs’ ‘AZE’

L329. Comma after ‘[33]’

Author Response

Dear Reviewer 1,

Thank you for your thorough review and your supportive response to our manuscript. We have incorporated each of your suggested edits and enhanced both the abstract and the introduction as recommended. The manuscript is better because of your input and we are grateful.

Onnie Byers

Reviewer 2 Report

A wonderful paper! Well written and stimulating. My comments are marginal really, but will hopefully be of interest to the authors.

Line 49: have geographical ranges outside of protected areas?

Line 82: surely the species of conservation concern should be considered as stakeholders in this system? Pop viability, ranging, and other “sound data” are insufficient; behaviour (movement, diet breadth / choice, reproductive choices, learning, intra and inter-species interactions and so on) must be placed into the model alongside “sound science” which is limited in its definition here. Note Table 2 mentions “biology” and needs to include Behaviour. Biology is fine (yes, encompasses behaviour) but for the general reader, and aiming for impact, this is too lose a term to encompass what actually makes species go extinct (it’s their behaviour in their habitats, interacting with human behaviour in their habitats).

Table 1: Consensus: it is worth noting that achieving consensus may be impossible, but enabling a stakeholder voice across all concerned individuals enables a form of “environmental and intergenerational justice”. It might be worth a mention as the recent UN declaration on Rights for Nature include a perspective on procedural rights for nature as a stakeholder.

The inclusion of Fig 2 is excellent; however the problem with the original study is that some species are “easier” than others to reduce declines (e.g. the fish in the study?), so is this generalisable? Other studies highlight the difficulty of determining conservation success over the long-term (many many recent papers: e.g. Adams et al 2019, or Akcakaya et al 2018 – the green list etc) I am sure the authors are more than aware of these issues, but conservation optimism may be a bit premature here.

Line 304: will require INVESTMENT. It is money that will make the development of these capacities successful. Not just funding for “conservation” as discussed below, but money to make the planning processes possible.

Author Response

Dear Reviewer #2,

Thank you so much for your enthusiastic, supportive review of our manuscript. Your comments were not only very helpful but also thought provoking! I hope we have addressed them adequately. Our paper will be better for it!

Onnie Byers

Reviewer 3 Report

To improve this paper, it must delve into the methods and original data from this study.

Author Response

Dear Reviewer #3,

Thank you for your feedback on our manuscript. We agree with you and the Academic Editor that the type of article is better described as a Perspective.

We appreciate your advice.  

Onnie Byers

Reviewer 4 Report

This is a very well written manuscript pointing to urgent conservation needs and suggesting required conservation actions.

This ms needs no substantial improvement.

I only suggest to mention in lines 47-48 ("...managing invasive species.") also disease outbreaks, as they play a substantial role in threatening particular vertebrate groups as for example chytridiomycosis in amphibians.

p. 9 line 267 and 268: one time you use i.e., and one time you use e.g.. Is this on purpose?

p. 11, line 341 -345: the quotation marks are not complete

p. 12, line 298: two words stuck together

page 9, paragraph “Rapid identification of species needing planning. Here it could be included that species who need intensive care (see Figure 3) can be prioritised by using the different immense databases available to us, such as ZIMS and others, in order to perform gap analyses which species are held and bred successfully, and which are highly threatened but do not have safety net populations yet.

 A citation here could be  

Krzikowski, M., Nguyen, T. Q., Pham, C. T., Rödder, D., Rauhaus, A., Le, M. D. & T. Ziegler (2022): Assessment of the threat status of the amphibians in Vietnam - Implementation of the One Plan Approach. – Nature Conservation, 49: 77-116.

Here, IUCN status was analysed, compared with microendemism to show current gaps in conservation, in concert with protected area coverage and ex situ conservation component. The microendemism analysis was performed to guide local institutions and stations where (and for which taxa) prioritized action is required. Okay, but this is only a suggestion, not a must.

In the end also a schema could be presented highlighting the most important actions / steps needed, to show most important steps of action in one diagram. But this is only a suggestion the authors could think about, this is not a must, for sure.

Happy to see that important work being published soon.

I don't want to be treated anonymous and my name can be shown to the authors.

Author Response

Dear Reviewer #4,

Thank you so much for your enthusiastic and supportive review of our manuscript. Your comments have helped to improve the paper and we are grateful!

Please see the attachment for our detailed response.

Onnie Byers

Round 2

Reviewer 3 Report

To the authors,

The new version clearly enunciates the manuscript's purpose, and the body is congruent with the outcomes. I only request to make minor changes. To do so, I have attached the paper with minor modifications. In the references, there is a misspelled surname.

Author Response

We are deeply grateful to Review 3 for supportive feedback and a thorough second review. We have revised as suggested, please see the attachment. Thanks again, Onnie

Specifically:

Line 27: One Plan Approach and Conservation Planning Specialist Group have been moved to the start of the keywords list

Table 2 (Preparing to Plan): comma added after 'rationale'

Line 167: comma added after 'gathering'

Line 342: comma added after 'confident'

Reference #3: corrected misspelling of 'Ceballos'
